# Exploration of the Adaptive Capacity of Residents of Remote Mountain Villages

**Shu-Hsun Chen \* and Bor-Wen Tsai**

Department of Geography, National Taiwan University, No. 1, Sec. 4, Roosevelt Rd., Taipei 10617, Taiwan; tsaibw@ntu.edu.tw
\* Correspondence: d98228001@ntu.edu.tw; Tel.: +886-0-2275-67189

**Abstract:** It is important to understand how residents in highly vulnerable natural and social environments, e.g., remote mountain villages, adapt to extreme climate shocks. Taking Shenmu Village in Xinyi Township, Nantou County, central Taiwan, as an example, this study examined the adaptive capacity of residents in this remote and mountainous area from the perspective of social capital and institutions. The empirical data for this study were collected from two sources: the Public Participation Geographic Information System Workshop and in-depth interviews with the residents of Shenmu Village. The results of the study reveal that the residents of Shenmu Village adopted agricultural adaptation strategies by switching crop types and utilizing diversified crop production spaces. Their adaptive capacities are based on mutual assistance and reciprocity, networks, local knowledge, mountain area and land management policies, and improvements in transportation and communication infrastructure in central Taiwan. This study can provide a reference for the sustainable development of remote mountain villages.

**Keywords:** adaptive capacity; agricultural adaptation; remote mountain village; Shenmu Village; social capital; social institutions

## 1. Introduction

Adaptation is a current topic of interest in the academic community. With the recent increase in the frequency of extreme climate events, the importance of the adaptation of people living in vulnerable environments has been highlighted. Mountainous areas play a major role in global ecological niches. Globally, mountains cover over 24% of the world's surface area and provide more than 80% of the global water supply. Approximately 12% of humans live in mountainous areas, and more than 50% of the world population directly or indirectly depend on mountains for their livelihood [1]. Half of the world's biodiversity hotspots are located in mountainous areas [2,3]. The natural resources and human activities in mountainous areas have a significant impact on mountain ecosystems. However, people living in mountainous areas often face greater natural and social vulnerability than those in flatland areas.

Taiwan, a country located in the eastern part of Asia, where Northeast Asia and Southeast Asia meet, is island-like with several mountains. Its mountainous areas, with altitudes above 1000 m, account for 39% of the total land area [4]. These areas are often characterized by steep terrain, high slopes, and fragile geological environments. Additionally, frequent earthquakes, heavy rainfall, and landslides often cause traffic disruptions, buried houses, and casualties [5]. Taiwan's unique geographical and climatic environment makes it imperative to understand and explore the challenges faced by residents of remote mountain villages with high vulnerability in adapting to extreme weather, which has become a global concern. Adger (2004) pointed out that adaptive capacity refers to the internal characteristics of a society (or system), which will affect its ability to face, deal with, and adapt to changes, in which social institutions and social capital play an

important role [6]. Pelling (2005) posits that social capital offers ways into understanding the role of fundamental social attributes that contribute toward building capacity for social collections and individuals to respond to climate [7]. From the perspectives of social capital and institutions, this study aimed to analyze the adaptive capacity of residents of remote mountain villages in overcoming their environmental vulnerability to achieve sustainable resource utilization.

This paper comprises six sections. After the Introduction, the concepts of adaptation in remote mountain villages, social capital, and social institutions are discussed in Section 2; the background information of Shenmu Village and local residents and research methods utilized for the study are provided in Section 3; Section 4 covers the analysis of adaptive capacities; Section 5 presents the Conclusion; and Section 6 summarizes future challenges and suggestions.

## 2. Concepts from the Literature

### 2.1. Adaptation and Adaptive Capacity

The term "adaptation" has recently been revived and discussed within the context of global environmental change, receiving international attention [8]. Adaptation (either from a biological/evolutionary or cultural point of view) emphasizes the survival of species (individuals) or groups when facing environmental change [9–14]. In the process of individual and group survival, cultural, institutional, and socio-economic contexts are profoundly shaped and adapted [15]. In the present study, we explored the adaptation strategies adopted by the residents of a remote mountain village in the face of environmental changes from the perspectives of social capital and institutions. It is also important to understand the local context of mountainous areas [16]. Actor-oriented analysis of adaptation views adaptation as a process of decision-making, responding to specific environmental stimuli, and emphasizing reducing vulnerability [17]. This paper defines adaptation as "the process by which actors respond continuously to changes in the external environment".

Adaptive capacity refers to the ability of a system to modify or change its characteristics or behavior so that it can better deal with existing or expected external pressures [6]. It is seen as a positive attribution of reduced vulnerability [18]. Understanding adaptive capacity implies knowledge of the universal capacities that exist in society for self-protection, collective action, and coping with stress [7]. This paper adopts the perspective of adaptive capacity under the study of vulnerability and regards adaptive capacity as a way to reduce vulnerabilities, emphasizing individuals' agency. The process of reducing vulnerability is adaptation.

The exploration of adaptive capacity focuses on local, community, and household scales, in which asset/capital, social, and biological factors are the main elements considered [19]. Asset-oriented analysis of adaptive capacity has been criticized for being ineffective in analyzing the dynamics of adaptive capacity and the power relations that determine adaptive capacity at local scales [20]. The local adaptive capacity (LAC) analysis framework proposed by the Africa Climate Change Resilience Alliance considers the specific and dynamic characteristics of adaptive capacity: asset/capital, institutions and entitlements, knowledge and information, and innovation. These four characteristics reflect high capacities. It will be explained as follows. Firstly, asset/capital refers to the availability of crucial resources that enable a system to respond and adapt to changing circumstances. Secondly, institutions and entitlements pertain to the presence of an appropriate and dynamic institutional framework that ensures equitable access to key assets and capital. Thirdly, knowledge and information denote the system's capacity to gather, analyze, and disseminate information and knowledge. Lastly, innovation implies that the system fosters an enabling environment that promotes experimentation, innovation, and the exploration of niche solutions to capitalize on emerging opportunities [20]. This study used the LAC framework to observe and analyze the adaptive capacity of remote mountain villages.

## 2.2. Social Capital and Social Institutions in Remote Mountain Villages

Remote mountain villages are located in mountainous areas and settlements formed due to mountain forest industry activities [21]. Mountain villages are characterized by an economy in which the livelihood of local residents is largely dependent on natural resources. The important developmental contexts regarding the mountain villages in Taiwan are the Han Chinese camphor makers entering the mountains and lumberjacks going up the mountains [22].

Social capital and institutions profoundly affect adaptive capacity [6]. Social capital represents the organizational characteristics of a society, including trust, norms, and networks [23] that accelerate the cooperative action of communities to boost the efficiency of society; these characteristics are regarded as "a set of networks, treaties, and flows". Social institutions contain rules of behavior for individuals and groups [24,25]. These norms are guided by several factors, including historical experience, war, and the environment [26]. Social capital is an important element in coping with climate variability and hazards in the present day [7,27]. It is represented by public goods, reducing transaction costs, and facilitating the exchange of resources and information, and improves our understanding of informal social relationships, giving meaning to trust and reciprocity [28,29]. Social capital and social institutions can enhance resilience in the context of resource-development livelihoods [30]. From the perspective of social capital and institutions, this paper explains why the residents of remote mountain villages are able to adapt to a highly vulnerable environment.

## 3. Case Study: Shenmu Village and the Hakka People in Shenmu Village

### 3.1. Background and History

Shenmu Village is a remote mountain village located in Xinyi Township, Nantou County, central Taiwan (Figure 1). The altitude of the village ranges from 1000 to 2000 m. The local rock structure is composed mainly of massive and sandy shale in the Heshe layer, and the loose structure of the sandy shale facilitates water seepage, forming groundwater channels and resulting in landslides [31]. The topographical characteristics of the area, coupled with high-intensity rainfall, have led to an annual average cumulative rainfall of 3447 mm over the past 15 years [32]. The geological characteristics and heavy rainfall in the rainy season render the region prone to landslide disasters. In 1996, heavy rainfall due to Typhoon Herb caused a serious landslide disaster in the area, and Shenmu Village became a reputed disaster region in Taiwan. In August 2009, Typhoon Morakot-induced mudslides hit the region again, causing the destruction of many homes and bridges and damage to farming, forestry, fishery, animal husbandry, and civil facilities in Nantou County, with an estimated cost of TWD 5.9163 billion [33].

Local geological characteristics and rainfall have caused recurrent landslide disasters. Because the average altitude of the area is over 1000 m, the annual average temperature is considerably lower than that of flatland areas. The abundant rainfall in this area facilitates the production of out-of-season vegetables (also known as summer vegetables) with a high economic value. The quality of vegetables produced in summer in flatland areas is poor due to high temperatures and pests, whereas the Shenmu region can produce high-quality vegetables that cannot be produced in flatland areas, creating a major niche in the local agricultural market.

Shenmu Village is located in the upper reaches of the Chen Youlan River watershed, an important river in central Taiwan, and is the uppermost settlement in this area. The inhabitants of the region mainly comprise the Hakka ethnic group among the Han Chinese, whereas the middle and lower reaches of the Chen Youlan River watershed are dotted with aboriginal (Bunun and Tsou) and Han Chinese settlements. The diverse ethnic composition of the region demonstrates the unique distribution of ethnic groups in the central mountainous area of Taiwan.

Initially, the region was used by the indigenous Tsou people for hunting. During the Japanese colonial period (circa 1910), the Hakka immigrants who originally lived in the Miaoli, Hsinchu, and Taoyuan areas were recruited by the colonial government to assist the

government in developing forest resources in the central mountainous area of Taiwan. Initially, the main job of these Hakka immigrants was to cut down camphor trees to extract camphor oil. Owing to the rich forest resources, the region was designated as an experimental forest for the Imperial University during the Japanese occupation. The main purposes of setting up the experimental forest were to investigate the forestry conditions inside and outside the area and the demand for forest products and to conduct camphor production tests.

After the end of World War II in 1945, which ended the Japanese colonial occupation of Taiwan, the experimental forest was temporarily taken over by the Forestry Bureau, and the National Taiwan University Experimental Forest was established in 1948 to manage the original forest, which includes the three townships of Lugu, Shuili, and Xinyi located in the Chen Youlan River watershed. Shenmu Village is located in forest compartments 28, 29, and 31 of the Gaoyue Camp in the National Taiwan University Experimental Forest. Local residents can only lease land from the government, and they have the right to use the land without ownership.

The Hakka people entered the Shenmu region during the Japanese colonial period to develop forestry resources and formed the settlement that exists today. After the Japanese colonial government ended its rule in Taiwan in 1945 and the Kuomintang Government came to Taiwan, local residents participated in a decades-long deforestation effort between the 1960s and 1980s, in conjunction with the government's stand conservation program, with approximately 3000 hectares being deforested. The village was originally formed as a result of the exploitation of nearby forest resources, but as the national forestry policy shifted from forest resource exploitation to conservation, local residents turned to growing summer vegetables and carried out forest specialty activities on the side.

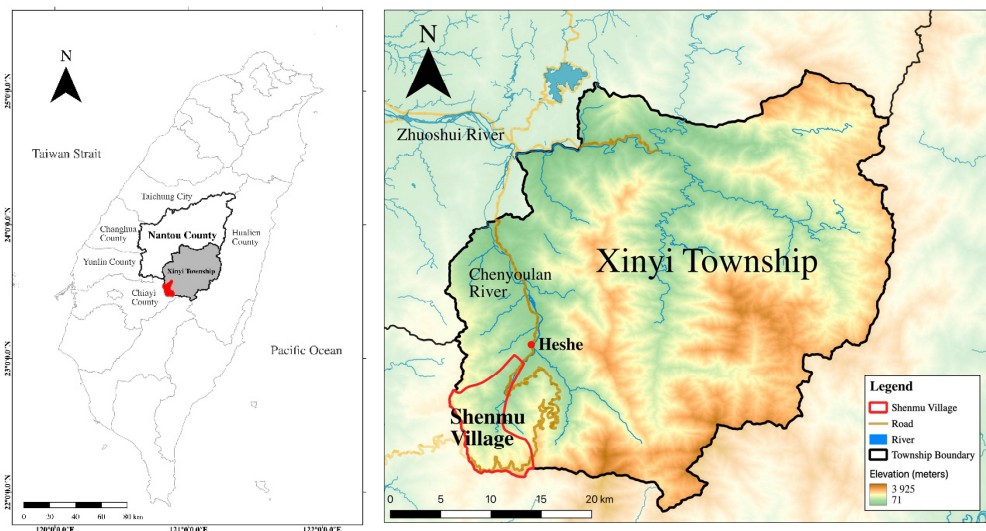

**Figure 1.** Location of study site, Shenmu Village, a remote mountain village located in Xinyi Township, Nantou County, central Taiwan.

### 3.2. Methods

Data were collected through in-depth interviews and the Public Participation Geographic Information System (PPGIS) workshop. The interviewer followed with the interests concerning how local residents have adaptive capacity to face environmental change and the role of social capital and institutions. The PPGIS workshop focused on the needs and abilities of specific users, so that relatively disadvantaged groups in society could effectively utilize geographical information to engage in public participation and assert their rights [34].

A PPGIS workshop was conducted on 27 November 2010, at Shenmu Elementary School, which brought together 29 local residents and included 19 male and 10 female residents who participated in this workshop (Figure 2). The age distribution of participating residents ranged from 40 to 80 years old. In this way, the workshop could collect information

on long-term environmental and local livelihood changes. The facilitator used various pictures such as old photos of the Shenmu area, topographic maps from the Japanese colonial period, and aerial photos taken between 1980s to 2000s. These old image data might help participants recall past living experiences in this place. Then, facilitators and participants used Google Maps to identify important points and environmental changes (Figure 3). The topic of discussion was important historical and disaster events affecting the area, and critical points and spatial information were marked on the map. The PPGIS workshop allowed the researchers to grasp the developmental context of the village and the major policies affecting the area, which served as a source of analysis for social institutions. Through in-depth interviews with local farmers, the study examined the social capital of local residents within the context of agricultural adaptation.

This paper adopts a case study approach. According to Yin, a case study typically adopts a triangulation approach and uses multiple sources of evidence [35]. This paper collects multiple materials, including in-depth interviews, PPGIS workshops, and literatures to explore and understand the issues of environmental adaptation of residents in remote mountain villages and local context.

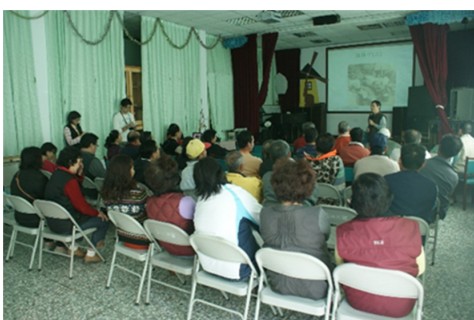

**Figure 2.** The facilitator and participants at the PPGIS workshop (Taken on 27 November 2010).

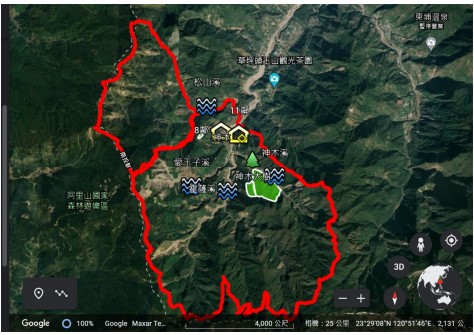

**Figure 3.** PPGIS workshop mapping results. PPGIS mapping identified the main rivers, spiritual symbols (such as the giant tree), and the logging boundaries within Shenmu Village, as indicated by the red frame. (Taken on 27 November 2010).

## 4. The Analysis of Adaptive Capacity

Why do Shenmu Village residents have adaptive capacities that reduce their environmental vulnerabilities? The following presents an analysis of the adaptive capacities of residents of Shenmu Village through the framework of local adaptive capacities, considering three dimensions: assets/capital, knowledge, and institutions.

### 4.1. Demonstration of Social Capital

#### 4.1.1. Mutual Assistance and Reciprocity of Local Residents

Mutual assistance and reciprocity among local residents are reflected in several contexts, such as the loan of production tools, the exchange of labor among households during the busy season, and the exchange and sharing of crop cultivation techniques and other information.

As an example, take the extraction of citronella oil in the 1960s. After harvesting, citronella needs to be refined to produce citronella oil. Interviewee H1 (H1 interview transcript on 14 June 2013.) explained that to refine citronella into citronella oil, it was necessary to have equipment to "tsiam" (The Taiwanese word "tsiam" means "oil dripping out" (extrusion).) the citronella oil and make it in a citronella hut. However, not every household had a citronella hut that could "tsiam" citronella oil, and only those who produced more citronella oil would have a citronella hut. Those who produced less citronella oil would borrow citronella huts from villagers to "tsiam" the oil. The borrowing of citronella huts to "tsiam" the oil and the villagers who borrowed them saying "I will 'tsiam' more oil for you" in return reflect the spirit of mutual assistance and reciprocity among local residents. Residents solve the problem of insufficient production tools through the loan of production tools and the feedback of borrowers.

Summer vegetable cultivation has become an important source of income for local residents (Figures 4–7). During the busy summer season when vegetables are grown, local residents also support each other by exchanging labor forces. This village is located upstream of the Chenyoulan River. Geographical isolation is more frequent among the mixed aboriginal and Han people settlements located in the middle and lower watershed. In these communities, Han people do not ask aboriginal people to help in the busy season; they rely on their neighborhood's labor.

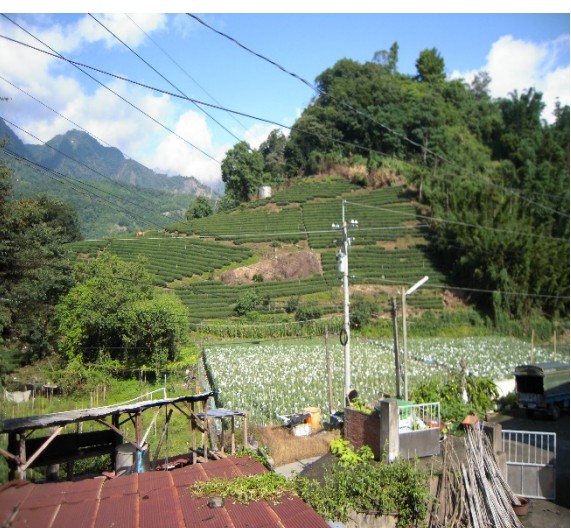

**Figure 4.** Summer vegetable planting and landscape in the Shenmu region (Taken on 9 February 2011).

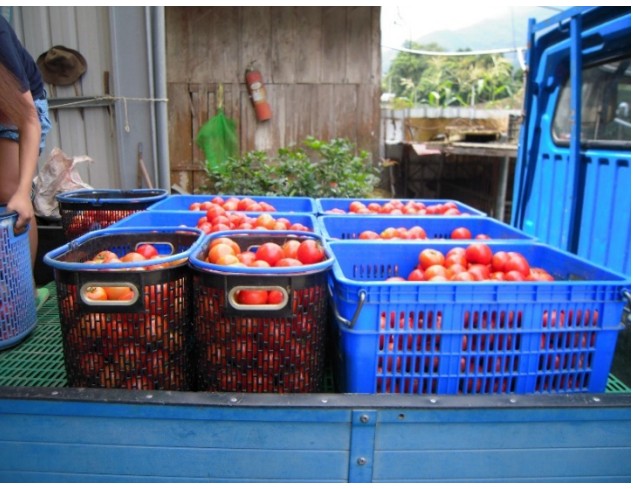

**Figure 5.** Summer vegetables in the Shenmu region—tomatoes (Taken on 9 February 2011).

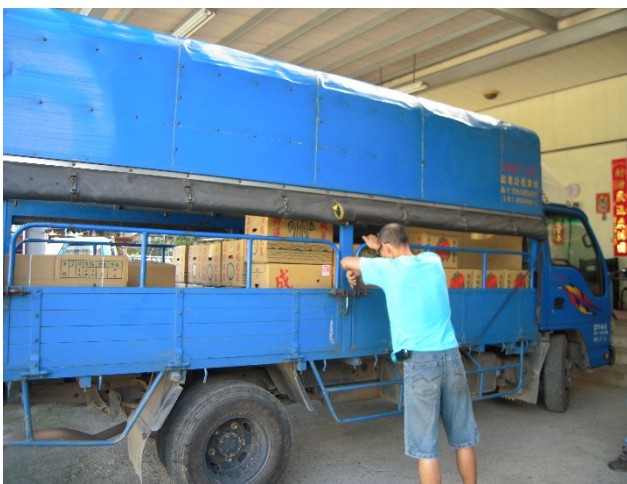

**Figure 6.** The transportation of vegetables from Shenmu Village to Yunlin Xiluo is facilitated through the use of vegetable carts. (Taken on 9 February 2011).

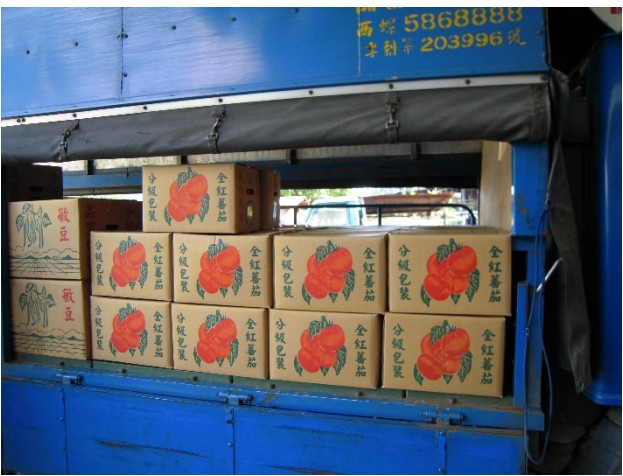

**Figure 7.** Crops (green beans and tomatoes) ready for shipment) (Taken on 9 February 2011).

Local residents also share agricultural information and technologies; for this reason, they can grow different crops to meet market demand, alongside planting castor, citronella, fruit trees, mushroom, tea, and summer vegetable. Notably, the interviewees highlighted that although farmers share crop grafting techniques, pest control is a matter of their own skill. The Hakka community characteristically helps each other and believes that everyone profits from the sharing of knowledge. Pest control technology is another matter, and it is considered "proprietary" in a way; such knowledge is not so generously shared. After Typhoon Morak in 2009, local residents lost sections of riverbed that they had used for a long time. Residents turned to new land belonging to aboriginal tribes in the lower–middle area of the Chenyoulan River. They also exchanged news of land to seek and rent to avoid being deceived.

4.1.2. Local and Hakka Kinship Networks

Local and Hakka kinship networks are a primary source of information for Shenmu residents of remote mountain villages. Shenmu residents grow different crops at different times, and crop information exchange is a large element of the local and Hakka kinship networks in Miaoli, Hsinchu, and Taoyuan. The exchange of information between kin networks living in the Chenyoulan River watershed through marriage is also a major source of information exchange on new crop cultivation for local residents.

The interviewees (H1 interview transcript on 2 September 2010; H4 interview transcript on 12 February2011.) bought shiitake mushroom strains from the Hakka villages in Taoyuan Daxi and Miaoli Toufen through their interpersonal network in their hometown in Taoyuan for trial planting, and the dark nets needed to build shiitake mushroom huts were purchased from the neighboring village of Tongfu. Together with the network of relatives among the residents of the Shenmu region, Hakka women who had married outside the region in the 1970s brought their husbands back to the Shenmu settlement to grow mushrooms together, making mushroom an important crop in Shenmu at that time. The wood needed for mushroom cultivation was left behind after logging and was also purchased from the southern region in Taiwan. Mushroom huts were built around their homes. At that time, mushroom cultivation was closely linked to the logging work. When mushroom cultivation was in full swing, the number of students in Shenmu Elementary School exceeded 100 (H3 interview transcript on 11 February 2011), which indicates the prosperity gained from mushroom cultivation.

Interviewee H5 (H5 interview transcript on 15 June 2013.) explained that in the 1980s, when alpine tea was popular, his sister married in the nearby village of Caopingtou, and he went to Caopingtou to learn tea growing techniques and became the first person to grow tea in the Shenmu region. As a result, other people in the region started to grow tea.

During the period of martial law (1949–1987), people who lived in flatlands could not go to mountain areas arbitrarily. Therefore, the communication between residents in the Shenmu Village and the outside world is more complicated than in the flatlands. However, they went to Taoyuan, Hsinchu, and Miaoli and brought back information about crop planting or exchanged information with relatives who lived in the Chenyoulan River Basin. Local and Hakka kinship connections were thus important sources of information in this period.

In addition, Bamboo shoots are an important forest by-product. Harvesting and processing of bamboo shoots reflect the connection between livelihood and social capital/networks. Bamboo shoots must be processed after harvesting to preserve. They need to be peeled and cooked in a large pot, which is very time-consuming and labor-intensive. Residents must rely on the assistance of each household to complete this work.

*4.2. Accumulation of Local Knowledge*

Local knowledge refers to facts or information from specific places or place-based contexts. The local residents have lived here since the Japanese colonial period (1910). They have, at various stages, been engaged in camphor production, logging work, and planting various cash crops. They have thus accumulated a wealth of local knowledge.

The early techniques of planting castor and citronella and refining them into oil can be traced back to collecting camphor during the Japanese colonial period. Logging and the opening of the forest road during the government's stand conservation program meant that local people could obtain discarded cut logs from the mountains for growing shiitake mushrooms. The successful trial plantings initiated the boom of shiitake mushroom cultivation in the region.

A mastery and understanding of the local environment are also indicated in the ability of the local people to use diversified production spaces for growing crops. Interviewee H4 (H4 interview transcript on 12 February 2011.) mentioned that the soil in the riverbed is not fertile, being poor and rocky; the weather is extremely hot and humid in summer; and the soil is prone to bad fungus, which is not conducive to crop growth. In order to overcome the problem of poor soil and fungus in the riverbed, local people harvest fertile black soil from the Zhuoshui River watershed every year and transport it to their local riverbeds for soil improvement, and then remove the huge rocks from the riverbed via rock pulling to create a conducive environment for crop production.

Local knowledge also reflects on the use of diversified crop production spaces in the Shenmu region. Residents plant crops suitable for different spatial environments based on the characteristics of the space. This innovative space use demonstrates the shortage

of flat land available for cultivation and a strategy adopted to generate more diversified crop income. In addition to the vast areas of riverbed land used by local residents to grow summer vegetables before Typhoon Morak in August 2009, the land around the settlement, which they call the "garden", as well as the hillside land and the forest land outside the settlement, were all used by local residents to produce crops. As interviewee H9 (H9 interview transcript on 26 December 2014.) explained:

> *"Bamboo is planted where there is a slope, and Ficus awkeotsang is planted where there are trees. A garden is made in a flat place, and the old garden which was lost will not be planted again."*

The locals use a variety of spatial characteristics for crop production: the cultivated land close to the home is used for intensive farming (such as growing summer vegetables) that is labor-intensive, whereas bamboo shoots are produced between the home and the forestland, and the forestland far from the home is used for Ficus awkeotsang, which is an extensive crop that does not require constant care from farmers.

The production area for extensive crops is located in the forestland outside the settlement. These crops only need to be harvested periodically, and farmers do not need to expend a considerable amount of time and energy to take care of them. However, these crops are cultivated far from the settlement, and the investment in harvesting is higher. Other crops that farmers invest their energy in, besides the extensive forest specialties and the intensive summer vegetables, are fruit trees. Fruit trees, such as plum and pear trees, are usually planted near farmers' homes. Although it takes longer to harvest fruits than vegetables, they can be grown on nearby land and require little effort to manage compared to summer vegetables. The third type of crop, summer vegetables, is the most intensive and accounts for the highest cost of care, and its spatial distribution has ranged from riverbeds in the past to hillsides today. In particular, for summer vegetables grown in riverbeds, farmers have to spend considerable effort on land and soil preparation every year. Hillside land is a relatively safe production space compared to riverbed land, which is susceptible to landslides. The only drawback is that it is less convenient to obtain water, but it is now the main area used by local farmers to grow summer vegetables, and some hillside land is used for tea cultivation.

### 4.3. Social Institutions

Social institutions are presented at both local and national scale. At the local scale in Shenmu area, these institutions are represented by local trade organizations. For example, vegetables carts from Xiluo in western Taiwan come to Shenmu Village to purchase vegetables. This trading behavior is primarily based on trust in pricing, rather than official polices. Agricultural adaptive behavior of the residents in the Shenmu area not only reflects the adaptive capacity, but also the impact of national policies and the context of regional development. The discussion will primarily focus on national policies, including mountain control and deregulation, the opening of the new central crossing highway, and the preparation of communication infrastructure in central Taiwan, as well as land control policies.

### 4.3.1. Mountain Control and Deregulation

In the past, most discussion on mountain control policies during the martial law period has focused on their impact on the aboriginals [36], with less attention being paid to their impact on Han Chinese living in aboriginal townships. The residents of Shenmu living in the aboriginal township were also affected by these mountain control policies. The development of agriculture in the Shenmu region was affected by mountain control policies and deregulation as well as the opening of the new, central, cross-island highway.

Mountain control policies started with the "Regulations for Strengthening the Security of Mountainous Areas in Taiwan Province" promulgated by the Ministry of National Defense in 1949, followed by the "Regulations for Controlling the Entry of National Army Troops into Mountainous Areas in Taiwan Province" promulgated by the Taiwan Defense

Command in 1950, the "Regulations for Controlling the Entry of Outsiders into Mountainous Areas in Taiwan Province during Martial Law" promulgated by the Ministry of National Defense in 1952, and the "Regulations for Controlling Mountainous Areas in Taiwan Province during Martial Law" issued by the Ministry of National Defense in 1965.

Article 1 of the "Regulations for Controlling Mountainous Areas in Taiwan Province during Martial Law" states the reason for the regulation of mountainous areas: "The regulations for controlling mountainous areas in Taiwan Province during Martial Law are established in accordance with Article 11 of the Martial Law to ensure the security of mountainous areas and to protect the interests of the people in the mountains". Article 2 specifies mountain control areas in Taiwan (During the martial law period, there were 30 mountain control areas in 12 counties in Taiwan, including Nantou County's mountain control areas: Ren'ai Township and Xinyi Township.) and the mountain control areas in Nantou County, including the Ren'ai and Xinyi townships. In the present study, the removal of the mountain checkpoints in Xinyi Township and Shenmu is considered as a reference point in the discussion. Currently, some areas in Xinyi Township have not yet been released from mountain control. (According to the United Daily News 18 February 2020, Heping District in Taichung City announced the lifting of the mountain control, while Xinyi Township in Nantou County has not yet been lifted due to local consensus and administrative procedures. Source: https://udn.com/news/story/7325/4351627, accessed on 16 March 2021).

The mountain checkpoints acted as an intermediate between the flatland and the controlled area. Upon entering Xinyi Township, one would encounter a checkpoint in the mountains and, upon entering Shenmu Village, another checkpoint in Shenmu. Thus, people or goods had to pass through two checkpoints to enter Shenmu Village. The checkpoints into Xinyi Township and Shenmu Village have since been removed; however, the Shenmu region was highly controlled during the mountain control period. During the nearly 40 years of mountain control, the residents in the Shenmu region and the local economy were connected to the outside economic market for crops.

Although access to the mountains was severely restricted during the control period, it did not affect the ability of Shenmu residents to transport their crops to outside markets (H1 interview transcript on 16 February 2020), especially to central distribution centers such as Heshe (also known as Tongfu), and the Alishan area, which is only a short distance away. The Tsou people were relocated to Wangmei Village (Jiumei) in the middle reaches of the Chenyoulan River watershed during Japanese rule. They had formerly lived in the "Fandi" (aboriginal area), which was originally used for hunting by the indigenous Tsou people. Contact with the indigenous people was inevitable after the Hakka people entered the area to collect camphor. According to local residents, the early Tsou aboriginals were allowed to carry hunting rifles. The conflict between local residents and the Tsou entering the Shenmu region for hunting was alleviated by mountain control. The mountain control measures were designed to prevent defectors/subversive elements from fleeing into the mountains to hide, and the 1954 Gao Yisheng incident in the Alishan region had a strong deterrent effect on the indigenous people and made the spatial separation between the indigenous peoples and the Han Chinese more distinct. The local Hakka Han people living in the mountainous countryside had a positive experience with mountain control:

> *"During the mountain control period, I felt safe in my life, the security was good, and I did not feel inconvenienced."* (H1 interview transcript on 16 February 2020)

The relationship between local residents and the indigenous people was more relaxed during the mountain control period. The security of the mountain area was safeguarded by measures to control the people entering the mountains.

### 4.3.2. The Opening of the New, Central, Cross-Island Highway and the Preparation of Communication Infrastructure in Central Taiwan

The opening of the new, central, cross-island highway and the Chi-Chi earthquake (921) led to the improvement in the communication and network infrastructure in the central

region, which produced another impact on external transportation and summer vegetable farming in the Shenmu region. This reduced the time taken by farmers to transport their crops to the field and increased the ability of local farmers to monitor market prices for their crops through communication networks. With the improvement in transportation and communication infrastructures in 1999 after the 921 earthquake, the entire central region was provided with a physical basis for neo-liberal development [37], making it easier for local residents to access information and move around.

In the 1970s, the telecommunications industry was deregulated, and the trend of liberalizing telecommunications in Europe and the United States began with control switching from the State to private companies. In 1987, the Ministry of Transportation and Communications (MOTC) promoted the liberalization of Taiwan's telecommunications in response to the deregulation and the implementation of the national policy of internationalization and liberalization. By the end of 1997, private mobile phone operators (2G) began official operations, and mobile phones quickly became an indispensable and important communication tool in people's daily life. The total number of mobile phone users increased dramatically from 1.94 million at the end of 1986 to 19.88 million in 1994, and the penetration rate increased from 6.88% at the end of 1986 to 87.29% at the end of 1994, a more than 12-fold increase. At the local level, the internet access rate per 100 households in Nantou County has already reached 37.78%, and it shows a trend of year-on-year growth, from 125.97 mobile phones per 100 households in 2000 to 174.55 mobile phones per 100 households in 2004. From 1996 to 2004, the number of cable TV channels per 100 households increased from 64.97 to 69.31 [38].

Due to the popularity of the Internet and communication devices such as cell phones, local farmers used the Internet to check the prices of fruits and vegetables on the market, kept track of the prices of their produce on the market, and contacted the "vendors" who came to collect their produce through cell phones. In this way, farmers administered a high degree of control over their own crop prices. Interviewee H2 (H2 Interview Transcript on 16 June 2013), a 70-year-old farmer who is an active cell phone user, stated:

> *"Currently, the information is readily available, and it is easy for local people to receive the market price of fruits and vegetables in Taipei. Sometimes the dealer's bid is not very high, so they just give him a few boxes. If the price is low, and if the dealer is interested in buying, they will raise the price."*

With the improvement in communication infrastructure in the central region after the 921 earthquake, access to personal communication devices and the Internet in general became more accessible.

### 4.3.3. Land Control Policy

Local residents were able to take advantage of a diverse range of crop production spaces thanks to several forestry policies promoted by local and forestry authorities at the time, including the availability of riverbeds for farming and allowing hillsides to be cleared and planted. During the period when summer vegetable farming on riverbed land was popular, i.e., before Typhoon Morakot in August 2009, local residents still maintained riverbed land cultivation, and they leased riverbed land from NTU Experimental Forest, and were able to rent a larger area of land for summer vegetable farming than today. After Typhoon Morakot, the management of the riverbed land was transferred from NTU Experimental Forest to the River Bureau, because the soil and rock flow damaged the farming environment of the riverbed land, and the River Bureau no longer leased the riverbed land to local residents.

Governmental policy is a major factor affecting the development of local agriculture. Opening up the riverbed land for summer vegetable farming and encouraging farmers to develop hillside land have contributed to the cultivation of summer vegetables in the area.

## 5. Conclusions

The paper presents the following features of residents' adaptation: adaptation should have a temporal and locational context; adaptation is a scale-dependent issue; infrastructure can enhance motivation; and learning is an important factor in sustainable adaptation.

### 5.1. Adaptation in a Temporal and Locational Context

This study was conducted to identify the adaptation strategies of residents of Shenmu Village in central Taiwan in response to environmental changes. The results of the study indicate that adaptation occurs in a specific temporal and locational context. By understanding the developmental context of the area over time, it is possible to explore the cumulative impact of social institutions and capital on the adaptive behavior of Shenmu residents. Current adaptive behavior is based on previous life experience. North [25], in his 1993 Nobel Prize in Economics lecture, mentioned that:

> "*In the context of time, it is not only about the current experience and learning, but also the accumulation of experiences rooted in the culture of past generations.*"

Time is the most important medium for developing adaptations, and it presents changes in the process of adaptation. The formation of Shenmu Village, including the first generation of residents who moved from the Hakka region of northern Taiwan for camphor production, has a century-long legacy. Through the Japanese colonial period when the area was used as an experimental forest, and after the arrival of the Kuomintang Government, in the special historical context of mountain control and forestry changes under the stand conservation program from the 1960s to 1970s, the agricultural adaptation demonstrated by the residents of the Shenmu region and the unique geographical location of the Shenmu region highlight the uniqueness and irrevocable adaptation of local residents. This uniqueness refers to the characteristics of ethnic groups, regions, and historical and cultural contexts. The adaptation behaviors developed under these contexts cannot be replicated by other people or regions.

### 5.2. Adaptation as a Scale-Dependent Issue

Scale has always been a concern for adaptation and adaptive capacity, which manifest in a scale-dependent relationship. Policies and systems at larger scales are bound to affect the adaptation of communities, households, and individuals at a smaller scale. This case study of the Shenmu region provides an empirical study of peoples' adaptation at the local scale, which is embedded in multiple scales of policy. The shift from forestry to agriculture in the Shenmu region reflects the response of the residents to national-scale land management policies, mountain control and deregulation, and infrastructure development. These complex and intertwined adaptation behaviors are also evident in the post-disaster reconstruction plan proposed by the State in the late 1990s, as the frequency of natural disasters has increased. Although the permanent housing provided a safe home for local residents, they chose to commute between the two places as a compromise in this system because of the limited arable land in the permanent housing sites and the fact that the natural environment of the permanent housing sites differs greatly from that of the old settlements in Shenmu. This shows that the adaptation strategy under the national scale cannot take into account the more detailed needs of the local community, so it is necessary to address these issues from perspective of local residents as well.

### 5.3. Transportation and Communication Infrastructure Enhances Motivation

In the early 1980s, the opening of the new, central, cross-island highway provided quicker access to the northern and western towns in the Shenmu region and the uppermost settlement of the Chenyoulan River. The opening of the new, central, cross-island highway facilitated the transport of the high-quality vegetables grown by the residents of the Shenmu region to the western markets in a shorter period of time. With the improvement of the road outside of the region, one method of local agricultural product transportation was taken up by fruit and vegetable "vendors" from the Xiluo region of Yunlin County, who

would drive their small trucks to local farmers' homes to pick up their crops and transport them to other places.

In 1999, following the 921 earthquake, with the construction and rapid development of transportation and communication facilities such as fast roads, the Internet, and cell phones, the material basis for the neo-liberalization of the Greater Taichung region was provided [39]. With the convenience of the Internet, local residents were able to inform themselves about the market price of vegetables without being exploited by traders. With summer vegetables as the mainstay of the local economy, the external environment (infrastructure, experimental forest, and non-compulsory land use control) also provided opportunities for local residents to adapt to market mechanisms.

*5.4. Learning as an Important Factor for Sustainable Adaptation*

From the perspective of social institutions and capital in understanding the adaptive capacity of the people in the Shenmu region, local people have been able to grow crops that are suitable for the local environment and meet the market demand by switching crops when this would be economically beneficial. Growing summer vegetables is a labor-intensive task. After harvesting one year's crop, the soil has to be re-plowed with an excavator and prepared for the following year. Only the type of crops planted will be adjusted: peppers may be planted this year, but beans may be grown next year. Interviewee H1 (H1 interview transcript on 14 June 2013) explained:

> *"Every year we plant different crops; we ask questions as we plant, and we don't plant whatever we want."*

Despite the fact that policy and market changes have provided residents with the conditions to produce more economically beneficial summer vegetables, the residents were willing to share information about crop cultivation with each other and to seek knowledge about crop cultivation from the Taichung Agricultural Improvement Farm to improve their skills. However, if they had lacked the spirit of learning, they would not have been able to master the techniques of planting seedlings, grafting, and working with different farming environments. As the local people say, "they keep looking for change in order to face the changes in the environment".

If the local people had not been able to learn new communication technologies and use cell phones and other communication media to inform themselves about the price of crops on the market and negotiate with traders, they would have been exploited by them. The effectiveness of this infrastructure is highlighted by the ability of the local population to learn. The infrastructure would not have been materially helpful had the local population lacked the ability to master communication tools.

This paper examines the adaptive capacity of residents in remote villages through the lens of social capital and social institutions. We believe that with mutual assistance and reciprocity, and with reciprocity and the network relationship of social capital, residents in Shenmu area can break through the limitations of geographical space. Residents exchange labor to solve the problem of insufficient manpower in households. Furthermore, they can obtain crop planting information and technology through cross-watershed information exchange. In this way, they can change crops according to market demand and produce high price crops. The resilience of the marginal Han community, as noted by Taiwanese scholars, is attributed to the kinship [40].

The social institutions are able to improve local residents' adaptive capacity. For this case, policies are considered restrictive factors that include policies of mountain and land control. However, these policies prevent local land loss and promote local residents to develop agricultural technology to produce crop. The aim of constructing transportation and communication infrastructure is to facilitate the residents' ability to access information about crop prices and to transport their produce to various destinations. This article presents a case study highlighting the adaptive capacity of residents of remote mountainous areas. The findings of this case study can potentially inform the government to develop sustainable management policies for vulnerable mountainous regions.

## 6. Challenges and Suggestions for Future Research

The residents of the Shenmu region were partially relocated to the Jiadongjiao area in Nantou City in September 2011 as a result of a government-initiated resettlement program after Typhoon Morakot. The residents who moved to the permanent houses were faced with a very different natural environment in their new settlement. Given the extreme differences in altitude, soil quality, temperature, and rainfall between the new settlement and the mountainous farming environment they were familiar with, local residents chose to travel between the new settlement and the old one to return to the farming land in the old Shenmu Village for farming. Therefore, future research could focus on two aspects. The first involves exploring whether the residents who travel from old settlements to the farm have an advantage in cultivating summer crops in the face of intensifying climate change, and how local residents have adapted to the climate and market changes. The second involves examining the residents who have moved to the new settlement of Shenmu in Nantou City and how they have adapted to the new settlement, e.g., whether they have developed new adaptation strategies and taken advantage of their social capital.

**Author Contributions:** Conceptualization, S.-H.C. and B.-W.T.; methodology, S.-H.C. and B.-W.T.; software, S.-H.C.; investigation, S.-H.C.; writing—original draft preparation, S.-H.C. and B.-W.T.; writing—review and editing, S.-H.C. and B.-W.T.; visualization, S.-H.C.; supervision, B.-W.T. All authors have read and agreed to the published version of the manuscript.

**Funding:** This research received no external funding.

**Institutional Review Board Statement:** Not applicable.

**Informed Consent Statement:** Informed consent was obtained from all subjects involved in the study.

**Data Availability Statement:** Data are not publicly available, through the data may be made available on request from the corresponding author.

**Conflicts of Interest:** The authors declare no conflict of interest.

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
