# Peer review of "Exploration of the Adaptive Capacity of Residents of Remote Mountain Villages"

_sustainability, doi:10.3390/su15075917_

Round 1

Reviewer 1 Report

A few small typos:

line 67: ” [15].It is ”.   A space is required after the period.

l. 68: ”Engle,2011.Understanding

l. 75 and 70: ” analysis [17].As-”, ” scales[18]. The”…   The font is not uniform.

l. 78: ” (LAC)analysis

l.100: ” [25].Social   and other…

Author Response

Thank you for your suggestions.

Point 1: line 67: ” [15].It is ”.   A space is required after the period.

Response 1: We have reflected this commend by pg.2, line.76.

Point 2: l. 68: ”(Engle,2011).Understanding”

Response 2: We have reflected this commend by pg.2, line.77.

Point 3: l. 75 and 70: ” analysis [17].As-”, ” scales[18]. The”…   The font is not uniform.

Response 3: We have reflected this commend by pg.2, line. 84.

Point 4: l. 78: ” (LAC)analysis”

Response 4: We have reflected this commend by pg.2, line. 78.

Reviewer 2 Report

The paper, after minor revision according to the attached report, deserve to be published, Then, the corresponding author have 4 working days to upload the revised version for the final revision.
Best regards.

Reviewer 3 Report

Thank you for inviting me to review the manuscript [sustainability-2279675] entitled Exploration on the adaptive capacity of residents in remote mountain village” in Taiwan.

Abstract – authors can include the practical and policy implications of the study.

Introduction – Authors need to give justification for focusing on social capital and institutions as opposed to other perspectives.

Methods – The methodology is too brief for readers to appreciate the results. To provide clarity and improve understanding of the results, authors need to elaborate more on the case study approach/protocols used as suggested by authors such as Yin and others.

Section 4 (Analysis of Adaptive Capacity)

·       Interview transcripts can be put in italics.

·       Authors cannot avoid discussing livelihood, it seems social capital/networks is around livelihood issues.

·       Social capital has not come out much instead livelihoods have been more pronounced.

·       Similarly government policy is discussed much more than institutions such as local authorities, trade association for example. Authors need to clarify which institutions affect residents and also give justification for excluding others.

Conclusion – The conclusion highlight four aspects that adaptation should have a (1) temporal and locational context; (2) adaptation is a scale-dependent issue; (3) infrastructure can enhance motivation; and (4) learning is an important factor in sustainable adaptation. This conclusion does not make reference to the objective of the study which was to “examine the adaptive capacity of residents in a remote and mountainous area from the perspective of social and institutions. The conclusion has not tied in the four aspects above to social capital and institutions. Authors should show how the objective of the study have been achieved. A research question or research questions are needed to guide the authors.

General – The paper needs editing, for example, in Line 264, verify whether it is long-tern or long-term.

Round 2

Reviewer 3 Report

My comments have been fully addressed. The manuscript has been sufficiently improved to warrant publication in Sustainability.